# Biomedical Alloys and Physical Surface Modifications: A Mini-Review

**DOI:** 10.3390/ma15010066

**Published:** 2021-12-22

**Authors:** Xinxin Yan, Wei Cao, Haohuan Li

**Affiliations:** 1Department of Orthopedics, Renmin Hospital, Wuhan University, Wuhan 430060, China; xinxinyan@whu.edu.cn; 2Key Laboratory of Artificial Micro- and Nano-Structures of Ministry of Education, School of Physics and Technology, Wuhan University, Wuhan 430072, China

**Keywords:** biomedical alloys, surface modifications, mechanical biocompatibility

## Abstract

Biomedical alloys are essential parts of modern biomedical applications. However, they cannot satisfy the increasing requirements for large-scale production owing to the degradation of metals. Physical surface modification could be an effective way to enhance their biofunctionality. The main goal of this review is to emphasize the importance of the physical surface modification of biomedical alloys. In this review, we compare the properties of several common biomedical alloys, including stainless steel, Co–Cr, and Ti alloys. Then, we introduce the principle and applications of some popular physical surface modifications, such as thermal spraying, glow discharge plasma, ion implantation, ultrasonic nanocrystal surface modification, and physical vapor deposition. The importance of physical surface modifications in improving the biofunctionality of biomedical alloys is revealed. Future studies could focus on the development of novel coating materials and the integration of various approaches.

## 1. Introduction

Biomaterials are currently widely used in biological systems for medical purposes [1] such as dental applications, surgery, and pharmaceutical. Some of these materials are already commercialized in applications related to tissue growth and drug delivery. We exhibit some applications of biomaterials in Figure 1. The specifics effects and applications are determined by biomaterial properties [2,3]. These days, the research direction is the targeted design and control of biomaterial properties to achieve specific biological responses. The most critical parameter of a biomaterial is its absolute non-toxicity. Biomaterial type should be chosen with care for a specific medical application. For example, for drug release, the biomaterials are typically based on novel polymers [4]; for dental implants and bone plates, biomedical alloys are the best choices [5].

About 80% of all materials used for bio-implants are biomedical alloys. As the world population age at a fast rate, the demand for biomedical alloys is increasing rapidly. The most popular biomedical alloys include stainless steel [6], Co–Cr [7], and Ti alloys [8]; Figure 2 and Table 1 compare the properties of these three groups. Other biomedical alloys, based on Mg, Fe, Ta, and Nb alloys, are not as widely used [9]. Generally, the biofunctionality and the biological and mechanical biocompatibility of the currently used biomedical alloys should be improved to satisfy the growing variety of medical applications. Numerous efforts to improve the mechanical biocompatibility of all these alloys, including the strength, ductility, wear resistance, toughness, and corrosion, have been published in the scientific literature [10,11,12,13].

**Figure 1 materials-15-00066-f001:**
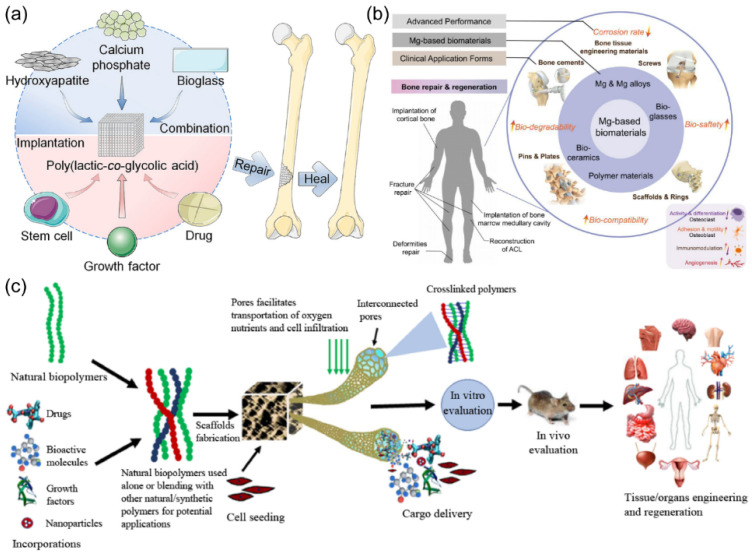
(**a**) Poly(lactic-co-glycolic acid)-based bone-substitute materials for bone repairing and healing. (**b**) Biomedical application of Mg-based biomaterials and their corresponding physiological processes. (**c**) The fabrication and role of biomaterials in the delivery of cells, bioactive molecules, growth factors, and drugs for tissue engineering applications. (Reproduced with permission from [3,14,15]. Copyright (2021), Elsevier.)

**Figure 2 materials-15-00066-f002:**
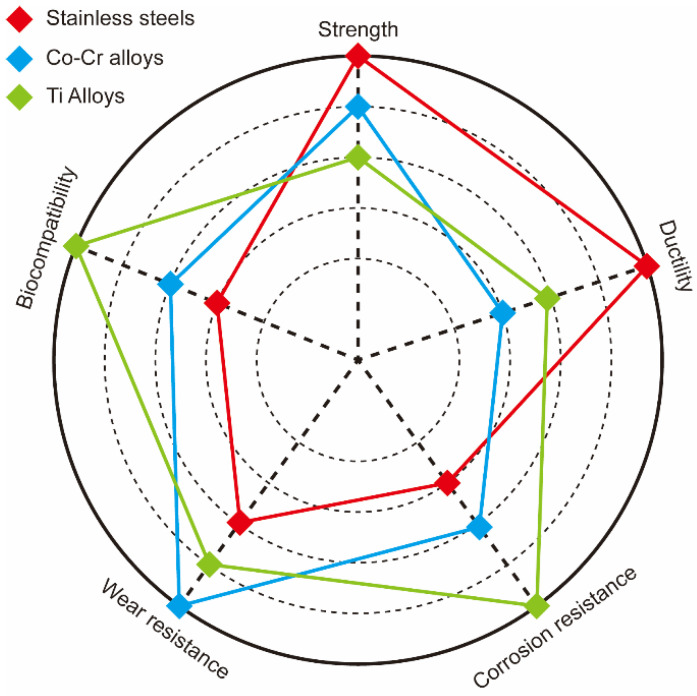
Comparison of strength, ductility, corrosion, and wear resistance as well as biocompatibility of stainless steel (red), Co–Cr (blue), and Ti (green) alloys.

The biofunctionality of biomedical alloys strongly depends on the properties of their superficial layers. Therefore, modification of these layers is a promising approach of tuning and improving various properties. Such surface modification methods, developed in recent decades [16,17], include mechanical, physical, chemical, and biochemical approaches [18]. The physical surface modification method involves a direct treatment of the superficial layers by thermal, kinetic, and electrical energy with almost no chemical modification to the original alloy matrix [19]. To fully understand the biofunctionality of biomedical alloys, a thorough analysis of their structures and physical surface modification mechanisms is essential.

Thus, this review first introduces common biomedical alloys (Co–Cr, Ti alloys, and stainless steels) and discusses their biomedical applications and potential improvements. Then, we present some physical surface modifications, including thermal spraying, glow discharge plasma, ion implantation, ultrasonic nanocrystal surface modifications, and physical vapor deposition. The same section discusses biomedical alloy applications modified by these methods. Finally, we compare various surface modifications methods and provide an outlook of the future progress in this field.

**Table 1 materials-15-00066-t001:** Summary of the properties, advantages, and disadvantages of the most common biomedical alloys.

Materials	Density	Tensile Strength	Elastic Modulus	Advantages	Disadvantages	Refs.
Stainless steels	~7500 kg/m^3^	~620 MPa	193~200 GPa	High strength, ductility	Harmful metal release, stress shielding	[20,21]
Co–Cr alloys	~10,000 kg/m^3^	~150 MPa	220~230 GPa	High strength and wear resistance	Harmful metal release	[22,23]
Ti Alloys	~4500 kg/m^3^	~240 MPa	55~114 GPa	High strength, corrosion resistance, biocompatibility, low elastic modulus	Stress shielding	[24,25]

## 2. Methods

All the authors of this study performed an electronic search set from 1991 to 2021 using Web of Science. The following keywords were selected individually or combined: biomedical application, biomedical alloy, stainless steel, Co–Cr alloy, Ti alloy, physical surface modifications, thermal spraying, glow discharge plasma, ion implantation, ultrasonic nanocrystal surface modification, and physical vapor deposition. Review articles and related research articles were also sources of references to locate other articles. About 15~20 articles were selected in each section. The inclusion criterion was that an article should contain biomedical alloys and physical surface modifications. Articles not providing enough information about biomedical application were excluded.

## 3. Biomedical Alloys

### 3.1. Stainless Steel

The application of stainless steel in biomedicine has the longest history among metallic biomaterials. Mature modern metallurgy successfully fabricates stainless steel with excellent properties. Stainless steel is a desirable material because its manufacturing is a mature, easy-to-perform, and inexpensive technology. Additionally, stainless steel possesses high corrosion resistance and mechanical strength. Stainless steels exhibit higher ductility and cyclic twist strength than Co–Cr and Ti alloys. Moreover, 316L stainless steel (Cr–Ni–Mo, “L” represents low carbon) is the most common one and is widely used for temporary and permanent implants [26] because it limits the formation of Cr–C and enhances the corrosion resistance [20].

As an orthopedic implant material, stainless steel should be non-ferromagnetic. Austenitic 316L stainless steels meet this requirement, and thus are used widely. Austenitic stainless steels can be specified as ASTM F138, ASTM F1586, or ASTM F2581 [27]. The ASTM F1586 and ASTM F2581 steels exhibit better corrosive resistance. Terada et al. [28] found that electrochemical treatment could effectively increase the corrosion resistance of austenitic stainless steels. However, these steels contained Ni, which is harmful and bio-incompatible [29]. Ni-free stainless steels are essential for health reasons. Yang and Ren [30] fabricated high-nitrogen nickel-free stainless steel with even better mechanical properties and superior biocompatibility. Talha et al. [31] further studied Ni-free N-rich austenitic steel (which they fabricated in an induction furnace), especially how the cold-working affected its mechanical behavior, and reported its excellent ductility and mechanical strength. The strain-induced martensitic transformation (SIMT) (austenite to martensite) in austenitic stainless steel could be caused by mechanical impacts, which proved to be favorable to cellular activity and hydrophilicity [32].

The usual way to treat stainless steels is through chemical methods, e.g., acid immersion and electrochemical anodizing. Stainless steel after acid immersion and anodizing treatment can exhibit the lowest thickness of the fibrous capsule membrane [33]. Yang et al. [34] investigated the effect of nitric acid passivation on high-nitrogen nickel-free stainless steels. The corrosion rate can be decreased by this passivation. Aguilar et al. [35] used the poly(caffeic acid) to coat the surface of stainless steel. Hsu et al. [36] applied an electrochemical anodizing method to modify the surface of 316L stainless steel. The electrochemical anodizing method can form a nanoporous oxide layer of Cr_2_O_3_ which can induce cell adhesion and promote bone formation. In addition to the above chemical methods, there are many other ways to improve the biomedical performance of stainless steel. Recent studies [37,38] proved that additive manufacturing can improve the charge transfer resistance and breakdown potential of 316 L stainless steel for clinical use. The cold deformation method can enhance the surface diffusion and corrosion resistance, which contributes to the passive films on the surface [39]. Moreover, Yang et al. [40] developed a simple and environmentally friendly water treatment to treat high nitrogen nickel-free stainless steels. This method increased passive films and allowed the corrosion rate of stainless steels to dramatically reduce to 1/20 of the untreated ones. Trzaskowska et al. [41] prepared a stainless steel coated with non-toxic organic materials by electropolymerization. These special organic coatings can fill surface scratches and reduce fibrinogen adsorption.

### 3.2. Co–Cr Alloys

Co–Cr alloys exhibit significantly higher wear resistance, heat resistance, and strength than Ti alloys and stainless steels. Co–Cr alloys also possess better corrosion resistance than 316L stainless steel. Therefore, Co–Cr alloys are commonly used to prepare bone substitutes, usually surrounded by Cl-rich body fluids, which could cause stress- and corrosion-related cracking if 316L stainless steel is used instead of Co–Cr alloy. When Co–Cr alloys are enriched with Cr, stable Cr_2_O_3_ film forms and protects the alloy from Cl-ion attacks [23,42]. Yamanaka et al. [24] even used a Co–Cr-based cast alloy for dental applications because of its high strength (comparable to wrought Co–Cr alloys) without ductility loss.

Similar to stainless steel, phase transition also exists in the Co–Cr alloy. Zhu et al. [43] researched the SIMT process in Co–Cr–W–Ni alloys and the process conforms to Schmid’s law. Ueki et al. [44] studied the precipitates that were induced during γ-ε phase transformation in Co–Cr–Mo alloys. The successful fabrication of Co–Cr–Mo alloy single crystals was first reported by Kaita et al. [45]. Mori et al. [46] proposed a novel approach to control the γ-ε SIMT in a hot-rolled biomedical Co–Cr–Mo alloy by manipulating the initial dislocation structures. The SIMT process was suppressed by the carbides.

The selective laser melting (SLM) technique is widely used for Co–Cr alloys. The SLM can relieve the stress concentrations and decrease the fatigue crack growth rate. A Co–Cr–Mo–W alloy treated by SLM exhibited longer fatigue life, higher tensile strength, and higher ductility than the untreated one [47]. Zhou et al. [48] produced Co–Cr devices by SLM which met the requirements for application as dental prostheses. Moreover, Co–Cr–Mo–W alloy treated by SLM showed better corrosion resistance, which was caused by effective micro-cathodes of precipitates [49].

Further improvements of the corrosion and wear resistances, as well as biocompatibility, could be achieved by introducing certain modifications to Co–Cr-based alloys. For example, Yamanaka et al. [50] developed a novel Co–Cr–W biomedical alloy for dental restorations with excellent fabrication and mechanical properties. Kheimehsari et al. [51] added hydroxyapatite (HA) coating to improve the corrosion resistance of Co–Cr-based implants. In this case, the thickness and sintering conditions of this HA-layer significantly affected the corrosion resistance of the Co–Cr-based implants. Shirdar et al. [52] applied HA/TiO_2_ coatings to Co–Cr–Mo alloys to enhance their mechanical and electrochemical properties. Sawangrat et al. [53] applied the harmonic structure design to synthesize new biomedical Co–Cr–Mo alloys with much better mechanical properties than the conventional biomedical alloys. Trimble et al. [54] developed a finite element model to predict the orthogonal forces of biomedical grade Co–Cr–Mo alloy and to reduce the number of machining tests. Migita et al. [55] used solid-binding peptides to improve the biocompatibility of the Co–Cr–Mo alloys. Yamanaka et al. [56] prepared Co–Cr–Mo alloy rods with a small diameter by hot-caliber rolling which exhibited high strength and durability. Mahajan and Sidhu [57] investigated the performance and biological responses of a Co–Cr alloy by an electrical discharge machining method, which assisted in improving the design precision for enhanced clinical performance.

### 3.3. Ti Alloys

Ti is a unique member of the biomedical alloy group as it possesses superior biocompatibility and complete inertness to the physiological environment, high corrosion resistance and strength, as well as a low elastic modulus. The density of Ti-based alloys is below that of stainless steels and Co–Cr alloys [25]. Some harmful elements could be released from stainless steel and Co–Cr-based alloys if they corrode or become damaged by wear [58]. However, such damage in Ti alloys can be entirely mitigated by forming very inert passivating TiO_2_ film [26]. Assis and Costa [59] confirmed electrochemically, by studying various Ti alloys, that high corrosion resistance correlates with the barrier layer presence and properties. The low elastic modulus of Ti alloys can also avoid the problem of stress shielding. Therefore, Ti alloys are often the best solution to solve a variety of biomedical problems. Commercially pure Ti is classified into four grades, G1 to G4. Acid etched Ti G4 exhibited better surface structures and mechanical properties, making it an ideal implant for dentistry [60,61].

Except for these commercially pure Ti, there are many Ti-based implants in the form of binary and multiple alloys. A typical representative of binary Ti alloys is Ti–Nb alloy. Porous Ti–Nb alloys synthesized by electro-deoxidation are proposed to be the best-suited candidate of the materials for biomedical applications [62]. Ibrahim et al. [63] also prepared porous Ti–Nb shape memory alloys by microwave sintering with the most uniform pore shape. Kuroda et al. [64] prepared Ti–Nb alloy via arc-melting which exhibited good mechanical properties and no cytotoxic effects. Apart from Ti–Nb alloys, there are also other binary Ti alloys, e.g., Ti–Ni alloys [65] and Ti–Fe alloys [66]. The additional metal content plays an important role in the biomedical application of these Ti alloys. The content of martensite phase in β-type Ti–Nb–Sn alloys decreases with the increase of Nb content, which has influence on the low elastic modulus. Maya et al. [67] found that Ti alloys with more Nb content exhibited excellent osteoinductive properties. [68] Qi et al. [66] studied the effect of the Fe content on the Ti–Fe biomedical alloy. The Ti-Fe alloy with the addition of about 5 wt% Fe content displayed excellent corrosion resistance.

Recently, a new variety of multiple Ti alloy was developed. Chui et al. [69] fabricated a series of novel β-type Ti–Zr–Nb–Mo where the corrosion resistance was mainly determined by the Mo content. Quadros et al. [70] prepared a Ti–Ta–Zr alloy with a low elastic modulus, excellent corrosion resistance, and no cytotoxic effects. Yılmaz et al. [71] produced porous Ti–Nb–Zr–Ta alloys by the space-holding method. These porous Ti–Nb–Zr–Ta alloys possessed suitable mechanical properties for hard tissue implants. A new β-type Ti–Nb–Mo biomedical alloy could exhibit a low elastic modulus, good wear resistance, an anti-wear capability, and a long service life [72]. Zhu et al. [73] reported a Ti-based bulk glassy alloy with great potential for biomedical application. Zhao et al. [74] fabricated another novel Ti alloy with outstanding corrosion resistance and superior mechanical biocompatibility.

Different preparation methods were developed to improve the biomedical performance of Ti-based alloys. Gao et al. [75] modified Ti-based alloy surfaces and enhanced their biocompatibility and stability even further. Xu et al. [76] used arc melting and graphite mold casting to prepare a series of Ti–Mo–Nb alloys, which simultaneously exhibited high strength and a low elastic modulus. Yang et al. [77] introduced gel-casting to obtain near-shape porous Ti alloys and that could even directly fabricate customized implants. The releasing of Cu ions is beneficial to lower infection incidences in Cu-bearing Ti alloys [78].

## 4. Physical Surface Modification of Biomedical Alloys

The degradation of the biomedical alloys based on stainless steel, Ti, and Co–Cr always starts on their surfaces. Thus, to improve or modify any properties of these materials (including corrosion and resistances as well as biocompatibility), a suitable surface modification approach must be used. These methods could be classified into treatment based on mechanical, physical, chemical, and biomedical techniques (see Figure 3). Below we discuss some of the physical surface modification methods.

### 4.1. Thermal Spraying

Thermal spray is an effective technology to improve wear resistance and biocompatibility through applying coatings. The thickness ranges from several microns to millimeters. The main methods of thermal spray are high velocity oxygen fuel spraying, flame spraying, plasma spraying, and so on. These methods can provide resistance to wear and corrosion which are favorable to biomedical applications [79,80,81,82,83]. Since these methods have similar principles, we mainly discuss plasma spraying in this review. Plasma spraying produces high temperatures (2700–11,700 °C). A typical energy source of this method is plasma arc heat. Metallic and ceramic powders can melt at such temperatures, which could be used to synthesize and apply a variety of coatings. The raw materials are heated or melted during the plasma spraying to coat the alloy surfaces at high speeds [84]. Liu and Ding [85] plasma-sprayed wollastonite coating and increased the bioactivity of Ti alloys. Zhou et al. [86] also used plasma-spraying and synthesized Ti alloys covered with thermal barrier coatings capable of withstanding very high temperatures. Sathish [87] developed a predictor of the tribological properties of 316L stainless steel treated by plasma nitriding, which is a significant achievement demonstrating the influence of physical surface modifications on the stainless steel properties.

Choosing the proper coatings is the key for the thermal spraying physical surface modification of biomedical alloys to target specific applications. For example, Ti alloys coated with specific composites showed better biocompatibility, wear resistance, and thermal stability than the uncoated ones [88]. Numerous coatings such as HA [89], CaO–SiO_2_ [90], CaSiO_3_ [91], TiO_2_ [92], and CaO–MgO–2SiO_2_ [93], were applied to biomedical alloys to realize or to improve their bioactivity. Among these coatings, HA coating is the most popular. Aruna et al. [94] prepared plasma sprayable grade HA powder which was non-toxic and beneficial for cells adherence. Pillai et al. [95] fabricated β-tricalcium phosphate (β-TCP) and HA/β-TCP composite coatings by a plasma spray process which could exhibit tunable solubility to satisfy specific biomedical applications. An advanced HA coating is fluoridated hydroxyapatite (FHA) coating. FHA coatings can be prepared by a suspension plasma spraying technique on a Ti substrate and greatly enhance the corrosion resistance [96].

However, the bond strength of biomedical composite alloys prepared by thermal spraying is weak due to the mismatched thermal expansion coefficients of the substrates and the coatings. The tensile forces often lead to cracking and/or delamination of such alloys [97]. To solve this problem, bond coatings, with the thermal expansion coefficient values between the substrates and original coatings, are used [97,98]. Moreover, there are some other improved plasma spray methods. Singh et al. [99] utilized atmospheric plasma spray (APS) to get the functionally gradient coating in a Ti–Al–V alloy which was assumed to promote early implant bonding with the host bone. Liu et al. [100] optimized the APS parameter to control the crystalline structure of HA coatings. Hameed et al. [101] developed a novel thermal spray process called axial suspension plasma spraying and the new method could lead to higher corrosion resistance.

### 4.2. Glow Discharge Plasma

Glow discharge plasma (categories of which include plasma surface modification [102], deposition [103], and polymerization [104]) also modifies the surfaces of biomaterials and the corresponding implants. It can clean and remove native oxides on the surfaces. Thus, glow discharge plasma can work at several nanometers. This approach is realized in an ultra-high vacuum and a high potential difference (~1 kV under the direct or alternating currents) between the corresponding electrodes [105] to produce an ionized gas, which determines the nature of the material being acquired.

Low-pressure plasma is a common treatment in glow discharge plasma. Truica-Marasescu and Wertheimer [106] used the low-pressure plasma polymerization of binary NH_3_/C_2_H_4_ mixtures to prepare N-rich organic coatings for biomedical applications. Yuvaraj et al. [107] modified the surface of carbon shell encapsulated HA by low-pressure plasma. The bioactivity of nanocarbon incorporated HA was enhanced. However, filamentary dielectric barrier discharge (FDBD) and atmospheric pressure glow discharge (APGD) have appeared as interesting alternatives. Sarra-Bournet et al. [108] discussed the potential of surface modifications realized with FDBD and APGD in different atmospheres (N_2_ + H_2_ and N_2_ + NH_3_ mixtures) which lead to very different surface chemistries.

Just as choosing the proper coatings in plasma spraying plays an important role, so too does the type of gas used in glow discharge plasma. The most common gas (air) has been used in glow discharge plasma for biomedical application [109]. Inert gas sources, e.g., Ar [110,111] and He [112] plasma, show a lot of advantages for biomedical applications: they are stable, low-cost, simple, and have enhanced plasma chemical activity. Apart from inert gas, Pandiyaraj et al. [113] investigated the surface properties of TiO_2_/PET films modified by O_2_ plasma which showed enhanced hydrophilicity, no cytotoxicity, and antibacterial activity. Jin et al. [114] co-implanted Zn and Ag into Ti alloys using plasma-immersion ion-implantation and obtained a material with excellent osteogenic and antibacterial ability, both of which are promising for applications related to orthopedic and dental implants.

Numerous papers have been published on the glow discharge plasma application to modify Ti alloys’ surfaces. Aronsson et al. [105] used glow discharge plasma with optimized parameters to remove surface contaminations and native oxides. Borgioli et al. [115] reported that glow discharge plasma was more efficient in hardening the Ti surface than simple annealing. Rossi et al. [116] prepared a nitrided Ti–Al–V alloy by glow discharge and reported a significantly better corrosion resistance than that of its unmodified counterparts. Plasma polymerization improves the immobilization of bioactive molecules and was implemented by Puleo et al. [117] to immobilize the bioactive molecules on a bioinert metal.

### 4.3. Ion Implantation

Ion implantation is considered an advanced physical surface modification method. The main application of ion implantation is improving resistance to wear, resistance, and fatigue. The depth range of ion implantation is about 1 μm. It is a low-temperature technique, during which ions of one element are accelerated into a solid target. Often, ion implantation is classified as a hybrid method consisting of a combination of beam-line and plasma immersion ion implantation approaches [118]. This method requires a vacuum to avoid contamination. Typical ion sources include ions of O, N, C, and metals. Tan and Crone [119] applied O-ion implantation to modify the surface of shape memory alloys to analyze their corrosion, wear, and biocompatibility. Li et al. [120] used ion-implantation to prepare Ti–Al–V alloys with Al in its oxidized state while V content could not be detected due to the stable outmost modified surface. The resulting alloy possessed significantly better wear and corrosion resistance as well as biocompatibility than its unmodified counterpart. However, ion implantation cannot be used for large-scale applications because of its high cost.

Research about ion implantation modifying surfaces has mainly concentrated on choosing the proper ion type. Ti-ion implantation could be utilized to modify pure Mg and improve the corrosion resistance and cytocompatibility [121]. Viviente et al. [122] applied C-ion implantation and increased the hardness of Ti alloys, while Rautray et al. [118] used N-ion implantation and increased the wear resistance of Ti–Al–V alloys. Dong et al. [123] applied Mn element implantation to the biomedical Mg surface and Mn ion and successfully modified the corrosion behavior. Jia et al. [124] conducted Zr-ion implantation to control the degradation of a biomedical Mg alloy. Different from these single elements, organic matter can also be chosen as an ion source. Wei et al. [125] introduced a method of carboxyl-ion (COOH^+^) implantation to reduce the degradation of Mg alloy and enhance the biofunctionality. Additionally, people have studied the collective effect of multiple ions. The incorporation of Ag and Cu ions increases the antibacterial activity of biomedical alloys [126,127]. Jörg et al. [128] modified Ti–Al–V surfaces by Ca- and P-ion implantation.

### 4.4. Ultrasonic Nanocrystal Surface Modification Techniques

A newly developed ultrasonic nanocrystal surface modification (UNSM) method is currently attracting increasing attention in the field of biomedical applications. It can improve the fatigue strength, wear, and corrosion resistance of alloys. The UNSM process can affect the mechanical properties in a range of up to approximately 750 μm in depth. UNSM is a method that applies ultrasonic impacts to generate nanostructured surfaces. During UNSM, the ultrasonic waves are passed through a tungsten carbide tip at high frequencies (~20 kHz) and transposed onto a sample surface (see Figure 4) [129]. The UNSM is unique in that it can produce precise hierarchical surface patterns by accurately adjusting the operation parameters. The temperature of UNSM is also an important parameter. UNSM at different temperatures was used to treat a Co–Cr–Mo alloy manufactured by SLM [130]. The effect of UNSM at high temperature is stronger than that at ambient temperature. UNSM improves the sample surface finish and decreases its porosity, which translates into improved biocompatibility, corrosion, and mechanical properties.

UNSM is also widely used for biomedical device treatments. Hou et al. [131] fabricated a hierarchical surface structure on Ti–Ni alloys via UNSM to improve corrosion resistance and hardness. Ye et al. [132] applied UNSM to prepare Ni–Ti alloys with an amorphous surface layer. The resulting material demonstrated high wear resistance and excellent biocompatibility. Kheradmandfard et al. [133,134] UNSM-treated Ti–Nb–Ta–Zr-based implants and created micropatterns on their surfaces, which benefited the implant’s bioactivity and bone regeneration performance. UNSM treatment also improved the wear resistance, biocompatibility [135], surface finish, hardness, and corrosion resistance [136] of Ti alloys while it decreased surface porosity. Hou et al. [137] also reported that UNSM significantly improved the hardness, yield stress, and wear resistance of Mg alloys as well as the mechanical and tribological properties of 316L stainless steel tubing [138] used in various biomedical applications. Ma et al. [139] used UNSM to treat the poor surface finish of 3D-printed metals and obtained significantly improved corrosion, wear, and fatigue resistance of the resulting 3D parts.

Advanced and innovative UNSM treatments were also reported in the literature. For example, Zhang et al. [140] developed electrically-assisted ultrasonication nanocrystal surface modification (EA-UNSM), capable of generating smoother surfaces and lower porosities than those of conventional UNSM. Amanov and Pyun [141] combined UNSM with the local heat treatment (LHT) and achieved a very hard Ti–6Al–4V alloy.

### 4.5. Physical Vapor Deposition

Physical vapor deposition (PVD) is also an effective way to enhance biofunctionality through modifying the surface of alloy implants. Similar to the methods discussed above, PVD protect implants from corrosive environments by applying coatings [142]. Generally, PVD coatings are dense and uniform but time-consuming. PVD can easily control the Ca/Pa ratio and structure, which makes Ca–Pa-based coasting widely used in PVD, such as HA [143], Si–HA [144], C–HA [145], and Mg–HA [146]. However, the deposition rate of PVD is not satisfactory for biomedical applications. The most common PVD processes are magnetron sputtering [147,148,149] and vacuum evaporation [150,151,152]. The main biomedical application of PVD is improving hardness, biocompatibility, wear, and corrosion resistance. The working depth range is up to approximately 4 μm. There are some examples for applying PVD on biomedical alloys. Ben and Khlifi [153] developed TiN biocompatible coatings on Ti alloys by cathodic arc evaporation vacuum deposition. To enhance corrosion resistance and biocompatibility, Li et al. [154] prepared MgF_2_ coatings on a MgCa alloy via the vacuum evaporation deposition method. Gonzalez et al. [148] deposited Ti–Nb coatings on 316L stainless steel substrates by magnetron sputtering. Kim et al. [155] modified the morphologies of HA coatings on Ti–Ta–Zr alloys with superior wettability by radio-frequency magnetron sputtering and a cyclic voltammetry.

## 5. Conclusions

Biomedical alloys are used to solve a variety of medical problems, especially those related to bones. Typical biomedical alloys include stainless steel, Co–Cr alloys, and Ti alloys. These biomedical alloys possess excellent corrosion resistance and mechanical properties, which makes them excellent materials for future biomedical uses. However, there are still problems on their surfaces during service. Developing surface treatment methods has attracted increasing attention. Many efforts have proved that physical surface modifications are effective and stable ways to enhance surface biofunctionality. Common physical surface modifications are thermal spraying, glow discharge plasma, ion implantation, ultrasonic nanocrystal surface modification, and physical vapor deposition. Although these methods have achieved certain results, further improvement is still necessary to satisfy the increasing surgical requirements. The main process of physical surface modifications is applying coatings on the substrates. Thus, investigating novel coating materials will be helpful in the future. Moreover, combining these physical surface modifications in a reasonable way can have them support each other and overcome their individual disadvantages. In summary, more efforts are needed to develop physical surface modifications on biomedical alloys for medical application.

## Figures and Tables

**Figure 3 materials-15-00066-f003:**
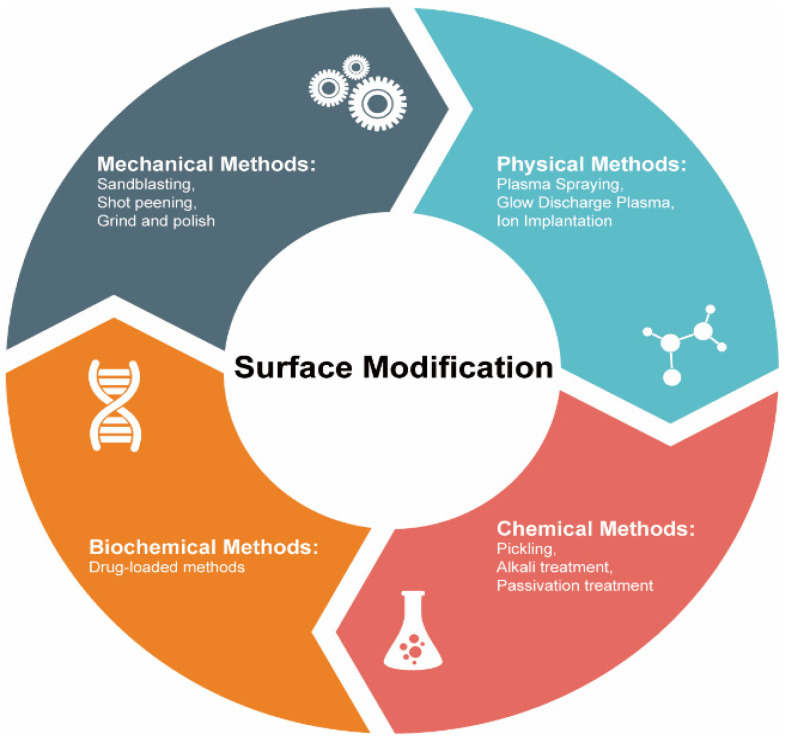
Commonly used methods to modify surfaces of biomedical alloys.

**Figure 4 materials-15-00066-f004:**
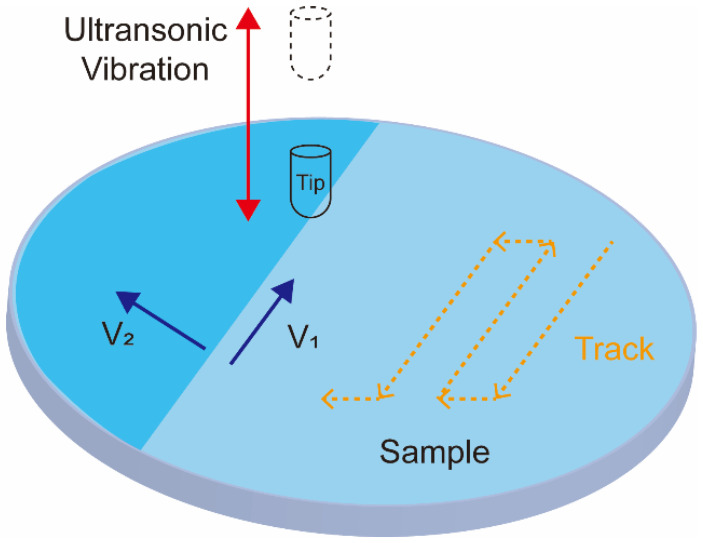
Schematics showing the ultrasonic nanocrystal surface modification process.

## Data Availability

The data provided in this study could be released upon reasonable request.

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
