# Peer review of "Biomedical Alloys and Physical Surface Modifications: A Mini-Review"

_materials, 2021, doi:10.3390/ma15010066_

Round 1

Reviewer 1 Report

The present manuscript entitled “Biomedical Alloys and Their Physical Surface Modifications: A Mini-review” falls under the scope area of Materials journal. Biomaterials, namely biomedical alloys are being extensively studied due to its great importance nowadays in fields like medicine, tissue engineering or prosthetics, and for this reason I am of the opinion that this review has reference value for related research.

Although, the authors should consider to revise the English, since parts of the manuscript are a bit confusing, namely sections 2.1 and 2.2. I suggest to use the same structure of ideas used in section 2.3 

Additionally, I address some comments to be considered:

1-In line 39 of the manuscript, if the intention is to mention literature refering improvments of mechanical properties and resistence to corrosion and wear of all the alloys mentioned (Mg, Fe, Ti, Co-Cr, etc.), then more references should be added here, since the one mentioned (ref. 10) only concerns to Ti alloys;

2-L99, organic instead of organical;

3-L102, "Co-Cr alloys exhibit wear resistance and strength (including wear resistance) significantly higher than Ti alloys and stainless steels." the text in brackets is repeated.

4-L107, "Cr2O3 film forms and protects the alloy from Cl- 107 ion attacks." is this Cr2O3 film non-toxic????

5-L112: SIMT stands for?

6-L154: "The acid etched Ti G4 performed better surface structures and mechanical properties which is ideal implant for dentistry. [53]", what the authors mean by “performed better surface structures”?

7-L247: "The most common gas air has been used in 247 glow discharge plasma for biomedical application." , “gas air” sounds strange please rephrase.

Author Response

We upload our response to the reviewer’s comments as a Word file.

Reviewer 2 Report

  1. What is the meaning in the title that “physical”? It is not needed in the title. Actually this term do not make a sense.
  2. Introduction section is very short and slim that authors need to explain their aims for this review in details.
  3. It would be beneficial to add and table to compare Co-Cr, SS and Ti alloys according to their properties such as density, tensile strength, elastic modulus etc.
  4. The surface modification techniques are explained very superficial, there should be more discussion of properties, studies, future of these methods.

Author Response

(The authors gave the same response as above.)

Reviewer 3 Report

The progress can be observed in the last decade in the area of biomaterials. Paper gives an overview of the main methods concerning the development of the physical surface modification techniques. The paper has appropriate and adequate references to related and previous work.

In my opinion this paper has some issues (presented below), which should be addressed by the authors.

  1. Abstract: Main conclusions from the review should be taken into account in this section.
  2. Tables and figures should be placed as near as possible to where the data is first referred to in the document.
  3. Methods of review according to the PRISMA (https://www.mdpi.com/journal/materials/instructions) must be specified in separate section. Each step of review process (i.e. search, screening, data extraction, etc.) should be described with sufficient detail to allow others to understand the assumptions of review process.
  4. The citation style needs to be revised throughout the manuscript. For example, reference no 20 should be referenced as Yang and Ren", not "Yang et al." (line 73)
  5. line 209: "HA". Each abbreviation must be defined the first time they appear in the main text.
  6. It is suggested to discuss other physical modification methods like flame spraying and high velocity oxygen fuel spraying. Moreover, the main methods of PVD, i.e. vacuum evaporation, ion plating, and sputter coating are omitted.
  7. The "Conclusions" section: This section is introduced rather abruptly, and it is quite unclear what is the main purpose and use of the paper. This section contains well-known statements that do not require a literature review. They can be found in any handbook. I recommend enlarging and synthesizing the conclusions based on the review results.
  8. The "Conclusions" section: The authors could mention future prospects of the research on physical surfaces modifications of biomaterials based on the conclusions obtained.

Author Response

(The authors gave the same response as above.)

Round 2

Reviewer 2 Report

The required revisions are done, thus the present state of the article is acceptable.

Reviewer 3 Report

The paper has been revised in detail according to the comments, and it has met the publishing requirements.

This manuscript is a resubmission of an earlier submission. The following is a list of the peer review reports and author responses from that submission.